# Autonomous Navigation of Unmanned Aircraft Using Space Target LOS Measurements and QLEKF

**DOI:** 10.3390/s22186992

**Published:** 2022-09-15

**Authors:** Kai Xiong, Peng Zhou, Chunling Wei

**Affiliations:** Science and Technology on Space Intelligent Control Laboratory, Beijing Institute of Control Engineering, Beijing 100094, China

**Keywords:** unmanned aircraft, autonomous navigation, LOS measurement, Q-learning, extended Kalman filter

## Abstract

An autonomous navigation method based on the fusion of INS (inertial navigation system) measurements with the line-of-sight (LOS) observations of space targets is presented for unmanned aircrafts. INS/GNSS (global navigation satellite system) integration is the conventional approach to achieving the long-term and high-precision navigation of unmanned aircrafts. However, the performance of INS/GNSS integrated navigation may be degraded gradually in a GNSS-denied environment. INS/CNS (celestial navigation system) integrated navigation has been developed as a supplement to the GNSS. A limitation of traditional INS/CNS integrated navigation is that the CNS is not efficient in suppressing the position error of the INS. To solve the abovementioned problems, we studied a novel integrated navigation method, where the position, velocity and attitude errors of the INS were corrected using a star camera mounted on the aircraft in order to observe the space targets whose absolute positions were available. Additionally, a QLEKF (Q-learning extended Kalman filter) is designed for the performance enhancement of the integrated navigation system. The effectiveness of the presented autonomous navigation method based on the star camera and the IMU (inertial measurement unit) is demonstrated via CRLB (Cramer–Rao lower bounds) analysis and numerical simulations.

## 1. Introduction

Recently, unmanned aircrafts flying at altitudes ranging from 20–100 km above sea level have received increasing interest in both the aeronautic and astronautic fields with respect to various applications [1,2,3]. As one of the key technologies of unmanned aircrafts, the precise navigation technique is essential for ensuring the feasibility and efficiency of the onboard guidance and control system [4,5,6].

Currently, the INS/GNSS integrated navigation system is widely used to provide information about the position, velocity and attitude of unmanned aircrafts, such as X-43A, X-51A and HTV-2. The INS/GNSS integration takes advantage of the consistently high accuracy of the GNSS and the short-term stability of the INS, which is promising for the achievement of precise navigation parameters [7,8,9]. However, the performance of INS/GNSS integrated navigation is easily affected by the disturbance of the GNSS signals. The use of INS alone is limited, as the navigation error increases with time, and it is sensitive to the initial errors [10,11].

To enhance the navigation performance, many researchers have focused on other autonomous navigation approaches [12,13,14,15,16]. A typical method is INS/CNS integration, where the accumulated errors of the gyroscopes in the IMU are compensated by the measurements of star sensors mounted on the aircraft [17,18,19]. Compared with the GNSS, one of the main characteristics of INS/CNS integrated navigation is that it relies on the optical signals from celestial bodies, such that the high-autonomous and anti-disturbance abilities of the navigation system are retained. The main limitation of traditional INS/CNS integrated navigation is that the position error accumulation caused by the bias of the accelerators in the IMU cannot be suppressed efficiently. Thus, it also suffers from an increasing number of errors over long flights. Achieving precise navigation information for unmanned aircrafts with a large flight envelope in the GNSS-denied environment remains a challenging problem.

In order to address this problem, this paper presents an INS/Vision integrated navigation method based on the star camera, which is mounted on the aircraft in order to measure the LOS vectors of space targets with the available position-related information, such as low-orbit satellites with a known ephemeris [20]. Different from the traditional star sensor, the star camera captures images of both the low-orbit satellites and the background stars in the field of view (FOV), and the LOS vectors of the satellites in the inertial frame can be extracted directly using the star catalogue and the least square (LS) algorithm. The current star cameras have the ability to observe low-orbit satellites and stars simultaneously during daytime when attached to unmanned aircrafts in near space. The measurements of the IMU and the star camera can be fused with a navigation filter in order to estimate the kinematic state of the aircraft.

The benefits of the presented navigation method are rooted in three aspects. Firstly, for unmanned aircraft in near space, the observation of the space targets, which are widely distributed within the celestial sphere, is less sensitive to the effect of clouds in comparison with landmark-based navigation [21,22]. Secondly, as it is easier to accurately measure the LOS vector of the space target than the direction vector of the Earth center using current technology, the presented method is promising for the achievement of a high rate of accuracy compared to navigation approaches based on the starlight and horizon reference [23,24]. Thirdly, both the position and the attitude information of the unmanned aircrafts can be measured with the star camera, which is able to observe the space target and the starlight simultaneously.

The main contributions of this work are as follows. Firstly, a novel autonomous navigation method is presented to determine the position, velocity and attitude of the unmanned aircrafts. Secondly, the theoretical accuracy of the presented method is analyzed through the calculation of the Cramer-Rao lower bounds (CRLB) based on the visibility analysis of the space targets. Thirdly, a Q-learning extended Kalman filter (QLEKF), which was designed to improve the navigation accuracy of space-target-based INS/Vision integration, is discussed.

This paper is divided into five sections. Following this introduction, the state and observation models of space-target-based INS/Vision integrated navigation are established in Section 2. Then, the method of the visibility analysis is outlined in Section 3. In addition, the QLEKF structure and an explanation of the learning process are given in Section 4. To demonstrate the performance of the presented method, a CRLB analysis and simulation study are described in Section 5. Finally, a brief conclusion is offered in Section 6.

## 2. System Model

### 2.1. Main Idea

The presented navigation method is based on the observation of a space target LOS vector, as shown in Figure 1. The LOS vector of the space target in the inertial frame is acquired through the processing of the image captured by the star camera, which contains both the space targets with a known ephemeris and the background stars in the star catalog. As the observation of the space target LOS vector can be implemented by the star camera alone, it does not depend on the misalignments between the different instruments, such that the possible sources of error are reduced. The observations of the space target LOS vector and the starlight vector are used to determine the position and attitude of the aircraft. It should be mentioned that the starlight vectors of the stars in the inertial frame can be calculated with an accuracy of less than 1 mas. The number and magnitude of the observed stars depends on the FOV and the sensitivity of the star camera. Generally, a star camera with an FOV on the order of $10^{{}^{\circ}}\times 10^{{}^{\circ}}$ and a sensitivity of about 8 Mv is feasible for navigation. 

The space-target-based INS/Vision integrated navigation system is mainly composed of three parts: the star camera, the IMU and the navigation filtering algorithm, as shown in Figure 2. The IMU is composed of three orthogonal accelerometers and three orthogonal gyroscopes, which are used to measure the angular rate and the specific force of the unmanned aircraft. 

The prediction of the aircraft’s position, velocity and attitude are obtained from the measurements taken by the gyroscopes and the accelerometers. The navigation filter, which is designed based on the state model and the observation model of the navigation system, is implemented in order to update the prediction of the position, velocity and attitude based on the measurements by the star camera. As the position, velocity and attitude of the unmanned aircraft are predicted using the IMU, with a high data update rate, the real-time accuracy can be guaranteed in the case when the exposure time of the star camera is recorded accurately. For the implementation of the navigation method, multiple space targets should be tracked by the ground stations, and the ephemeris of the space targets should be undated before the unmanned aircraft takes off. Then, the positions of the space targets can be obtained through orbital propagation during the flight of the aircraft.

### 2.2. Model for the Attitude Determination

The star camera and the gyroscopes are integrated in order to generate the attitude solution of unmanned aircrafts via the observation of the starlight so as to correct the attitude error and the gyroscope drift. The quaternion $q_{k}={\left[\begin{array}{cccc} q_{1k} & q_{2k} & q_{3k} & q_{4k} \end{array}\right]}^{T}$ is used to describe the attitude of the aircraft [25,26]. For the attitude determination system composed of the star camera and the gyroscopes, the state vector $x_{A,k}$ is considered as:(1)$$x_{A,k}={\left[\begin{array}{cc} \delta \rho_{k}^{T} & \delta b_{k}^{T} \end{array}\right]}^{T}$$
where $\delta b_{k}=b_{k}-{\hat{b}}_{k}$ is the estimation error of the gyroscope drift, $b_{k}$ and ${\hat{b}}_{k}$ are the gyroscope drift and its estimate, respectively, and the subscript *k* is the time label. $\delta \rho_{k}={\left[\begin{array}{ccc} \delta q_{1k} & \delta q_{2k} & \delta q_{3k} \end{array}\right]}^{T}$ is a part of the error quaternion $\delta q_{k}={\left[\begin{array}{cc} \delta \rho_{k}^{T} & \delta q_{4k} \end{array}\right]}^{T}$, where $\delta q_{4k}={\left(1-{∥\delta \rho_{k}∥}^{2}\right)}^{0.5}$. The operator ∥·∥ represent the Euclidean norm. The error quaternion $\delta q_{k}$ represents the difference between the estimated quaternion ${\hat{q}}_{k}={\left[\begin{array}{cccc} {\hat{q}}_{1k} & {\hat{q}}_{2k} & {\hat{q}}_{3k} & {\hat{q}}_{4k} \end{array}\right]}^{T}$ and the true quaternion $q_{k}$, which is expressed as:(2)$$\delta q_{k}=q_{k}⊗{\hat{q}}_{k}^{-1}$$
or
(3)$$q_{k}=\delta q_{k}⊗{\hat{q}}_{k}$$
The operator ⊗ represents the quaternion product, which is defined as:(4)$$q_{k}⊗{\hat{q}}_{k}^{-1}=\left[\begin{array}{cc} \Psi \left(q_{k}\right) & q_{k} \end{array}\right]{\hat{q}}_{k}^{-1}$$
with
(5)$${\hat{q}}_{k}^{-1}={\left[\begin{array}{cccc} -{\hat{q}}_{1k} & -{\hat{q}}_{2k} & -{\hat{q}}_{3k} & {\hat{q}}_{4k} \end{array}\right]}^{T}$$
(6)$$\Psi \left(q_{k}\right)=\left[\begin{array}{c} q_{4k}I_{3\times 3}-\left[\rho_{k}\times \right]\\ -\rho_{k}^{T} \end{array}\right]$$
where $\rho_{k}={\left[\begin{array}{ccc} q_{1k} & q_{2k} & q_{3k} \end{array}\right]}^{T}$, $\left[\rho_{k}\times \right]$ is a cross-product matrix defined by
(7)$$\left[\rho_{k}\times \right]=\left[\begin{array}{ccc} 0 & -q_{3k} & q_{2k}\\ q_{3k} & 0 & -q_{1k}\\ -q_{2k} & q_{1k} & 0 \end{array}\right]$$

The discrete-time state model of the integrated navigation system for the attitude determination is expressed as:(8)$$x_{A,k}=F_{A,k}x_{A,k-1}+w_{A,k}$$
with
(9)$$F_{A,k}=\left[\begin{array}{cc} I_{3\times 3}-\tau_{A}\left[\omega_{k}\times \right] & -\frac{\tau_{A}}{2}I_{3\times 3}\\ 0_{3\times 3} & I_{3\times 3} \end{array}\right]$$
where $\tau_{A}$ is the time step for the state prediction and $\omega_{k}={\left[\begin{array}{ccc} \omega_{xk} & \omega_{yk} & \omega_{zk} \end{array}\right]}^{T}$ is the estimated angular velocity, which can be obtained from the gyroscope measurements after the compensation for the drift. The denotation $\left[\omega_{k}\times \right]$ is formulated as:(10)$$\left[\omega_{k}\times \right]=\left[\begin{array}{ccc} 0 & -\omega_{zk} & \omega_{yk}\\ \omega_{zk} & 0 & -\omega_{xk}\\ -\omega_{yk} & \omega_{xk} & 0 \end{array}\right]$$

The process noise $w_{A,k}$ is modeled to obtain zero-mean Gaussian white noise with the covariance matrix $Q_{A,k}$.

For simplicity, as the star sensor is mounted on the aircraft, the aircraft body frame and the star camera frame are not distinguished in this paper. For the attitude determination, the starlight vector on the aircraft body frame is observed using the star camera. The measurement is considered as:(11)$$y_{A,k}=\left[\begin{array}{c} u_{Bk}^{\left(1\right)}-{\hat{u}}_{Bk}^{\left(1\right)}\\ u_{Bk}^{\left(2\right)}-{\hat{u}}_{Bk}^{\left(2\right)}\\ \vdots \\ u_{Bk}^{\left(N\right)}-{\hat{u}}_{Bk}^{\left(N\right)} \end{array}\right]$$
where $u_{Bk}^{\left(j\right)}$ and ${\hat{u}}_{Bk}^{\left(j\right)}={\left[\begin{array}{ccc} {\hat{u}}_{Bxk}^{\left(j\right)} & {\hat{u}}_{Byk}^{\left(j\right)} & {\hat{u}}_{Bzk}^{\left(j\right)} \end{array}\right]}^{T}$ (j=1,2,…,N) are the starlight vectors of the *j*-th star in the aircraft body frame and its estimate, while N is an integer, which denotes the number of visible stars in the FOV of the star camera. 

To construct an observation model which relates the state vector to the measurements, the following function of the quaternion is introduced:(12)$$u_{Bk}^{\left(j\right)}=A\left(q_{k}\right)u_{Ik}^{\left(j\right)}$$
where $u_{Ik}^{\left(j\right)}$ is the starlight vector of the *j*-th star in the inertial frame, which can be calculated according to the star catalog, while $A\left(q_{k}\right)$ is the attitude transformation matrix from the inertial frame to the aircraft body frame, which is formulated as:(13)$$A\left(q_{k}\right)=\left(q_{4k}^{2}-\rho_{k}^{T}\rho_{k}\right)I_{3\times 3}+2\rho_{k}\rho_{k}^{T}-2q_{4k}\left[\rho_{k}\times \right]$$
Accordingly, ${\hat{u}}_{Bk}^{\left(j\right)}$ is calculated as:(14)$${\hat{u}}_{Bk}^{\left(j\right)}=A\left({\hat{q}}_{k}\right)u_{Ik}^{\left(j\right)}$$
Substituting (3) into (12), we have:(15)$$\begin{array}{ll} u_{Bk}^{\left(j\right)} & =A\left(\delta q_{k}⊗{\hat{q}}_{k}\right)u_{Ik}^{\left(j\right)}\\ & =A\left(\delta q_{k}\right)A\left({\hat{q}}_{k}\right)u_{Ik}^{\left(j\right)} \end{array}$$
Inserting (14) into (15) yields:(16)$$u_{Bk}^{\left(j\right)}=A\left(\delta q_{k}\right){\hat{u}}_{Bk}^{\left(j\right)}$$

As the measurement accuracy of the star camera is on the order of a few arcsec, neglecting the high-order terms, the matrix $A\left(\delta q_{k}\right)$ is simplified as:(17)$$A\left(\delta q_{k}\right)=\left[\begin{array}{ccc} 1 & 2\delta q_{3k} & -2\delta q_{2k}\\ -2\delta q_{3k} & 1 & 2\delta q_{1k}\\ 2\delta q_{2k} & -2\delta q_{1k} & 1 \end{array}\right]$$
From (16) and (17), we obtain:(18)$$u_{Bk}^{\left(j\right)}-{\hat{u}}_{Bk}^{\left(j\right)}=2\left[{\hat{u}}_{Bk}^{\left(j\right)}\times \right]\delta q_{k}$$
where
(19)$$\left[{\hat{u}}_{Bk}^{\left(j\right)}\times \right]=\left[\begin{array}{ccc} 0 & -{\hat{u}}_{Bzk}^{\left(j\right)} & {\hat{u}}_{Byk}^{\left(j\right)}\\ {\hat{u}}_{Bzk}^{\left(j\right)} & 0 & -{\hat{u}}_{Bxk}^{\left(j\right)}\\ -{\hat{u}}_{Byk}^{\left(j\right)} & {\hat{u}}_{Bxk}^{\left(j\right)} & 0 \end{array}\right]$$

According to (11) and (18), for synchronously observed multiple stars, the measurement model is expressed as:(20)$$y_{A,k}=H_{A,k}x_{A,k}+v_{A,k}$$
where
(21)$$H_{A,k}=\left[\begin{array}{cc} 2\left[{\hat{u}}_{Bk}^{\left(1\right)}\times \right] & 0_{3\times 3}\\ 2\left[{\hat{u}}_{Bk}^{\left(2\right)}\times \right] & 0_{3\times 3}\\ \vdots & \vdots \\ 2\left[{\hat{u}}_{Bk}^{\left(N\right)}\times \right] & 0_{3\times 3} \end{array}\right]$$

The measurement noise $v_{A,k}$ is modeled as Gaussian white noise with the zero mean and covariance matrix $R_{A,k}$.

Using the system model shown in (8) and measurement model shown in (20), it is convenient to design a Kalman filter (KF) in order to estimate the state vector $x_{A,k}$ by means of the measurement $y_{A,k}$. Then, the estimates of $\delta \rho_{k}$ and $\delta b_{k}$ are used to correct the attitude quaternion ${\hat{q}}_{k}$ and gyroscope drift ${\hat{b}}_{k}$. The procedure of the attitude determination using the Kalman filtering algorithm is similar to that in [27], and it is omitted here for simplicity.

### 2.3. Model for the Position Estimation

In space-target-based INS/Vision integration, the space target LOS vectors observed using the star camera are used to correct the INS position and velocity errors in the inertial frame. The state vector for the position estimation consists of position vector $r_{k}={\left[\begin{array}{ccc} r_{xk} & r_{yk} & r_{zk} \end{array}\right]}^{T}$, velocity vector $v_{k}$ and accelerometer bias $\nabla_{k}$, which is expressed as:(22)$$x_{P,k}={\left[\begin{array}{ccc} r_{k}^{T} & v_{k}^{T} & \nabla_{k}^{T} \end{array}\right]}^{T}$$

The state model is established to describe the dynamics of the state vector, which is represented as:(23)$$x_{P,k}=f\left(x_{P,k-1}\right)+w_{P,k}$$
where $w_{P,k}$ is the process noise with the covariance matrix $Q_{P,k}$. The discretized nonlinear function f(·) is formulated as:(24)$$f\left(x_{P,k}\right)=\left[\begin{array}{c} r_{k}\\ v_{k}\\ \nabla_{k} \end{array}\right]+\left[\begin{array}{c} v_{k}\\ A^{T}\left(q_{k}\right)\left(f_{Bk}+\nabla_{k}\right)+g\left(x_{P,k}\right)\\ 0 \end{array}\right]\tau_{P}$$
where $\tau_{P}$ is the time step for the state prediction, $q_{k}$ can be obtained through the method shown in the previous sub-section, $f_{Bk}$ is the output of the accelerometers in the aircraft body frame, and $g\left(x_{P,k}\right)$ is a function used to describe the gravitational acceleration, which can be written as:(25)$$g\left(x_{P,k}\right)=-\frac{\mu_{E}r_{k}}{∥r_{k}∥{}^{3}}+p\left(r_{k}\right)$$
where $\mu_{E}$ is the gravitational constant of Earth and the function $p\left(r_{k}\right)$ encapsulates the perturbation accelerations of the aircraft other than those that are cause by the two-body gravitational acceleration. The Jacobian matrix of the state model is derived as:(26)$$F_{P,k}=\frac{\partial f\left(x_{P,k}\right)}{\partial x_{P,k}}=I_{9\times 9}+\Phi_{P,k}\tau_{P}$$
with
(27)$$\Phi_{P,k}=\left[\begin{array}{ccc} 0_{3\times 3} & I_{3\times 3} & 0_{3\times 3}\\ \frac{\partial g\left(x_{P,k}\right)}{\partial x_{P,k}} & 0_{3\times 3} & A^{T}\left(q_{k}\right)\\ 0_{3\times 3} & 0_{3\times 3} & 0_{3\times 3} \end{array}\right]$$
Let
(28)$$\frac{\partial g\left(x_{P,k}\right)}{\partial x_{P,k}}=\left[\begin{array}{ccc} \Phi_{P,k}\left(4,1\right) & \Phi_{P,k}\left(4,2\right) & \Phi_{P,k}\left(4,3\right)\\ \Phi_{P,k}\left(5,1\right) & \Phi_{P,k}\left(5,2\right) & \Phi_{P,k}\left(5,3\right)\\ \Phi_{P,k}\left(6,1\right) & \Phi_{P,k}\left(6,2\right) & \Phi_{P,k}\left(6,3\right) \end{array}\right]$$

For simplicity, only the two-body gravitational acceleration is taken into consideration. The elements in the matrix shown in (28) are written as:(29)$$\Phi_{P,k}\left(4,1\right)=-\frac{\mu_{e}}{∥r_{k}∥{}^{3}}\left(1-\frac{3r_{xk}^{2}}{∥r_{k}∥{}^{2}}\right)$$
(30)$$\Phi_{P,k}\left(4,2\right)=\frac{3\mu_{e}r_{xk}r_{yk}}{{∥r_{k}∥}^{5}}$$
(31)$$\Phi_{P,k}\left(4,3\right)=\frac{3\mu_{e}r_{xk}r_{zk}}{∥r_{k}∥{}^{5}}$$
(32)$$\Phi_{P,k}\left(5,1\right)=\Phi_{P,k}\left(4,2\right)$$
(33)$$\Phi_{P,k}\left(5,2\right)=-\frac{\mu_{e}}{∥r_{k}∥{}^{3}}\left(1-\frac{3r_{yk}^{2}}{∥r_{k}∥{}^{2}}\right)$$
(34)$$\Phi_{P,k}\left(5,3\right)=\frac{3\mu_{e}r_{yk}r_{zk}}{∥r_{k}∥{}^{5}}$$
(35)$$\Phi_{P,k}\left(6,1\right)=\Phi_{P,k}\left(4,3\right)$$
(36)$$\Phi_{P,k}\left(6,2\right)=\Phi_{P,k}\left(5,3\right)$$
(37)$$\Phi_{P,k}\left(6,3\right)=-\frac{\mu_{e}}{∥r_{k}∥{}^{3}}\left(1-\frac{3r_{zk}^{2}}{∥r_{k}∥{}^{2}}\right)$$

The space target LOS vector in the inertial frame is used as the measurement for the position estimation. Accordingly, the measurement model is expressed as:(38)$$y_{P,k}^{\left(i\right)}=h^{\left(i\right)}\left(x_{P,k}\right)+v_{P,k}^{\left(i\right)}$$
with
(39)$$h^{\left(i\right)}\left(x_{P,k}\right)=\frac{r_{Tk}^{\left(i\right)}-r_{k}}{∥r_{Tk}^{\left(i\right)}-r_{k}∥}$$
where $y_{P,k}^{\left(i\right)}$ is the measurement of the *i*-th space target LOS vector, $r_{Tk}^{\left(i\right)}$ is the position vector of the *i*-th space target, which can be calculated according to the known ephemeris and orbit propagation, and $v_{P,k}^{\left(i\right)}$ is the measurement noise with the covariance matrix $R_{P,k}^{\left(i\right)}$. The corresponding Jacobian matrix is:(40)$$\begin{array}{c} H_{P,k}^{\left(i\right)}=\frac{\partial h^{\left(i\right)}\left(x_{P,k}\right)}{\partial x_{P,k}}\\ =\left[\begin{array}{ccc} -\frac{1}{∥r_{Tk}^{\left(i\right)}-r_{k}∥}I_{3\times 3}+\frac{\left(r_{Tk}^{\left(i\right)}-r_{k}\right){\left(r_{Tk}^{\left(i\right)}-r_{k}\right)}^{T}}{{∥r_{Tk}^{\left(i\right)}-r_{k}∥}^{3}} & 0_{3\times 3} & 0_{3\times 3} \end{array}\right] \end{array}$$

According to the state model shown in (23) and the measurement model shown in (38), the extended Kalman filter (EKF) can be designed to fuse the accelerometer and star camera data. Furthermore, an advanced filtering algorithm, which is presented in Section 4, can be used to enhance the navigation performance.

## 3. Principle of Visibility Analysis

The visibility analysis of the space targets is the basis for the performance evaluation of the integrated navigation method. In practice, only the visible space targets can be observed with the star camera in order to provide valuable navigation-related information. The user should confirm whether certain space targets are visible to the aircraft before the implementation of the navigation scheme. To facilitate the application, certain principles used to analyze the visibility of the space targets are described in this section. 

First, for a space target to be visible, its apparent magnitude should be sufficient so as to be detected with the star camera, i.e.,
(41)$$m_{T}^{\left(i\right)}<m_{0}$$
where $m_{T}^{\left(i\right)}$ is the apparent magnitude of the *i*-th space target and $m_{0}$ is a magnitude bound specified by the sensitivity of the star camera. To simplify the notation, the time label k is omitted in this section. The apparent magnitude $m_{T}^{\left(i\right)}$ can be calculated as:(42)$$m_{T}^{\left(i\right)}=M_{T}^{\left(i\right)}+2.5\log_{10}[\frac{{∥r_{Sun}-r_{T}^{\left(i\right)}∥}^{2}{∥r-r_{T}^{\left(i\right)}∥}^{2}}{d_{0}^{4}p\left(\xi^{\left(i\right)}\right)}]$$
where $M_{T}^{\left(i\right)}$ is the absolute magnitude of the space target, which is approximated by:(43)$$M_{T}^{\left(i\right)}=m_{Sun}-5\log_{10}\left(\frac{rd^{\left(i\right)}}{d_{0}}\sqrt{al^{\left(i\right)}}\right)$$
where $m_{Sun}\approx -27$ is the apparent magnitude of Sun, $rd^{\left(i\right)}$ is the observed body radius, $al^{\left(i\right)}$ is the body albedo of the space target and $d_{0}\approx 1.4959787\times 10^{8}\;\mathrm{km}$ is an astronomical unit. $r_{Sun}$ is the position vector of Sun. $p\left(\xi^{\left(i\right)}\right)$ is a function to describe the integration of the reflected light, which is approximated by:(44)$$p\left(\xi^{\left(i\right)}\right)=\frac{2}{3\pi }\left[\left(\pi -\xi^{\left(i\right)}\right)\cos\left(\xi^{\left(i\right)}\right)+\sin\left(\xi^{\left(i\right)}\right)\right]$$
where $\xi^{\left(i\right)}$ is the sun–target–aircraft angle of the *i*-th space target, which is expressed as:(45)$$\xi^{\left(i\right)}=\cos^{-1}\left[\frac{r_{Sun}-r_{T}^{\left(i\right)}}{∥r_{Sun}-r_{T}^{\left(i\right)}∥}\cdot \frac{r-r_{T}^{\left(i\right)}}{∥r-r_{T}^{\left(i\right)}∥}\right]$$

It describes the relative positions of the sun, the space target and the unmanned aircraft, as shown in Figure 3.

Secondly, a necessary condition for the visibility of the space target is that it is not hidden by the shadow of Earth, which can be judged with the following condition: (46)$$\psi^{\left(i\right)}<90{}^{\circ}+\beta^{\left(i\right)}$$
where $\psi^{\left(i\right)}$ is the Sun–Earth–target angle of the *i*-th target, which is expressed as:(47)$$\psi^{\left(i\right)}=\cos^{-1}\left(\frac{r_{T}^{\left(i\right)}}{∥r_{T}^{\left(i\right)}∥}\cdot \frac{r_{Sun}}{∥r_{Sun}∥}\right)$$

From Figure 4, the angle bound $\beta^{\left(i\right)}$ is calculated as:(48)$$\beta^{\left(i\right)}=\sin^{-1}\left[\frac{{\left({∥r_{T}^{\left(i\right)}∥}^{2}-R_{e}^{2}\right)}^{0.5}}{∥r_{T}^{\left(i\right)}∥}\right]$$
where $R_{e}$ is the Earth’s radius. The space target is illuminated by the Sun and can be seen as a candidate observation object when it is not within the shadow of the Earth.

Thirdly, for the space target to be visible, it must not be occulted by celestial bodies, such as the Earth, Sun and Moon. A space target is believed to be occulted when its LOS vector points to a celestial body. Generally, the LOS vector of a space target cannot be measured effectively by the star camera in this case, as the images of the space target and the celestial body overlap. For example, the relative positions of the space target, the aircraft and Earth are shown in Figure 5. 

From the figure, to judge whether the space target is occulted by the Earth, the following condition is given:(49)$$\theta^{\left(i\right)}>\theta_{0}$$
where $\theta^{\left(i\right)}$ is the Earth–aircraft–target angle, which is expressed as:(50)$$\theta^{\left(i\right)}=\cos^{-1}\left(\frac{r-r_{T}^{\left(i\right)}}{∥r-r_{T}^{\left(i\right)}∥}\cdot \frac{r}{∥r∥}\right)$$

The angle bound $\theta_{0}$ is calculated as:(51)$$\theta_{0}=\sin^{-1}\left(\frac{R_{e}}{∥r∥}\right)$$

Similar conditions can be derived to judge whether the space target is occulted by the Sun or Moon.

## 4. Q-Learning Extended Kalman Filter

### 4.1. Q-Learning for the Filter Design

In this section, we describe an advanced navigation filter that was designed to estimate the position and velocity of the unmanned aircraft using the IMU and the star camera mounted on the aircraft in cases where the GNSS is unavailable. The EKF is a classic algorithm used to fuse the measurements of different sensors. An important procedure for the optimization of the EKF is the fine-tuning of the filtering parameters, such as the noise covariance matrices. However, the pre-determined noise covariance matrices based on the prior information may be not appropriate for aircrafts during flight, which will deteriorate the state estimation. In order to address this problem, we attempted to tune the process noise covariance with the measurement data online using the Q-learning approach so as to achieve a higher accuracy of the navigation system.

The Q-learning is a kind of reinforcement learning (RL) approach for solving sequential decision problems, where the agent interacts with an initially unknown environment and optimizes its behavior in order to maximize its cumulative reward [28,29,30,31]. Q-learning has received considerable attention, with many successful applications in various fields, such as gaming, robots, industrial process control and network management. Figure 6 illustrates the main components of Q-learning. An agent takes action a in the environment described by the state s, which may cause the change in the environment’s state (from s to s′), and receives an immediate reward R(s,a) from the environment as a feedback on the action. The learning objective is to modify the action selection policy such that the cumulative reward, which is represented by the Q function $Q_{i}\left(s,a\right)$, is maximized. Q-learning has been studied as an important data-based approach for adaptive control and optimal filters and is preferred for practical systems with uncertain models [32,33,34].

In this study, to deal with the continuous state and action spaces, a Q-learning method with function approximation is adopted, where the Q function is expressed as:(52)$$Q_{i}\left(s,a\right)=\Phi^{T}\left(s,a\right)\Theta_{i}$$
with
(53)$$\Phi \left(s,a\right)=\left[\begin{array}{c} \phi_{1}\left(s,a\right)\\ \phi_{2}\left(s,a\right)\\ \vdots \\ \phi_{D}\left(s,a\right) \end{array}\right],\;\Theta_{i}=\left[\begin{array}{c} \theta_{1i}\\ \theta_{2i}\\ \vdots \\ \theta_{Di} \end{array}\right]$$
where $\phi_{d}\left(s,a\right)$ (d=1,2,…,D) are basis functions, $\theta_{di}$ (d=1,2,…,D) are weights and the subscript i is the number of iterations. In this paper, the basis function is chosen as:(54)$$\phi_{d}\left(s,a\right)=\frac{exp\left(-\left[\begin{array}{cc} s-\mu_{sd} & a-\mu_{ad} \end{array}\right]\left[\begin{array}{cc} \sigma_{sd}^{-2} & \\ & \sigma_{ad}^{-2} \end{array}\right]\left[\begin{array}{c} s-\mu_{sd}\\ a-\mu_{ad} \end{array}\right]\right)}{{\left\{\sum_{\tau =1}^{D}exp\left(-2\left[\begin{array}{cc} s-\mu_{s\tau } & a-\mu_{a\tau } \end{array}\right]\left[\begin{array}{cc} \sigma_{s\tau }^{-2} & \\ & \sigma_{a\tau }^{-2} \end{array}\right]\left[\begin{array}{c} s-\mu_{s\tau }\\ a-\mu_{a\tau } \end{array}\right]\right)\right\}}^{0.5}}$$
where $\mu_{sd}$, $\mu_{ad}$, $\sigma_{sd}$ and $\sigma_{ad}$ are the parameters of the basis function. It should be mentioned that the choice of the basis function is determined by the user, and a poor choice may lead to a poor filtering performance. In this study, the basis function is designed with the structure similar to that in [35], such that the appropriate action can be activated. The incremental weight update rule of Q-learning is given by:(55)$$\Theta_{i+1}=\Theta_{i}+\alpha \left[R\left(s,a\right)+\gamma \underset{a^{\prime }}{\max}\left(\Phi^{T}\left(s^{\prime },a^{\prime }\right)\Theta_{i}\right)-\Phi^{T}\left(s,a\right)\Theta_{i}\right]\Phi \left(s,a\right)$$
where 0≤α<1 is the learning rate, 0<γ≤1 is the discount factor and α and γ are the tuning parameters. It is easy to observe from the previous equations that:(56)$$\Phi^{T}\left(s,a\right)\Theta_{i+1}=\Phi^{T}\left(s,a\right)\Theta_{i}+\alpha \left[R\left(s,a\right)+\gamma \underset{a^{\prime }}{\max}\left(\Phi^{T}\left(s^{\prime },a^{\prime }\right)\Theta_{i}\right)-\Phi^{T}\left(s,a\right)\Theta_{i}\right]\Phi^{T}\left(s,a\right)\Phi \left(s,a\right)$$
or
(57)$$Q_{i+1}\left(s,a\right)=Q_{i}\left(s,a\right)+\alpha \left[R\left(s,a\right)+\gamma \underset{a^{\prime }}{\max}Q_{i}\left(s^{\prime },a^{\prime }\right)-Q_{i}\left(s,a\right)\right]$$

The convergence of the iterative Q function is studied in the following theorem, which can help readers to comprehend the key idea of the learning procedure. 

**Theorem** **1:**
*Considering the iterative Q function sequence*

$\left\{Q_{i}\left(s,a\right),i=1,2,\ldots \right\}$

*, if the initial Q function*

$Q_{0}\left(s,a\right)$

*satisfies the following condition:*

(58)
$$0\leq Q_{0}\left(s,a\right)\leq R\left(s,a\right)+\gamma \underset{a^{\prime }}{\max}Q_{0}\left(s^{\prime },a^{\prime }\right)$$

*and there exist the constants*

$\bar{\delta }\geq 1\geq \underline{\delta }\geq 0$

*and*

λ≥0

*, such that the following inequalities hold:*

(59)
$$\underline{\delta }{\bar{Q}}_{\infty }\left(s,a\right)\leq Q_{0}\left(s,a\right)\leq \bar{\delta }{\bar{Q}}_{\infty }\left(s,a\right)$$


(60)
$$\lambda R\left(s,a\right)\geq \underset{a^{\prime }}{\max}{\bar{Q}}_{\infty }\left(s^{\prime },a^{\prime }\right)$$

*where*

(61)
$${\bar{Q}}_{\infty }\left(s,a\right)=R\left(s,a\right)+\underset{a^{\prime }}{\max}{\bar{Q}}_{\infty }\left(s^{\prime },a^{\prime }\right)$$

*then the iterative Q function*

$Q_{i}\left(s,a\right)$

*converges to the optimal Q function*

$Q^{*}\left(s,a\right)$

*as*

i→∞

*, i.e.,*

(62)
$$\underset{i\to \infty }{\lim}Q_{i}\left(s,a\right)=Q^{*}\left(s,a\right)$$

*where the function*

$Q^{*}\left(s,a\right)$

*is formulated as:*

(63)
$$Q^{*}\left(s,a\right)=R\left(s,a\right)+\gamma \underset{a^{\prime }}{\max}Q^{*}\left(s^{\prime },a^{\prime }\right)$$



The proof of the theorem is provided in Appendix A. It shows that, under certain conditions, the iterative Q function $Q_{i}\left(s,a\right)$ is convergent to the optimal Q function $Q^{*}\left(s,a\right)$, which represents the maximum cumulative reward. Generally, if the constants $\bar{\delta }$ and λ are sufficiently large, and $\underline{\delta }$ is sufficiently small, the conditions shown in (59) and (60) can be satisfied.

### 4.2. Filter Algorithm

Based on the system model for space-target-based INS/Vision integrated navigation, we designed the QLEKF algorithm, where the Q-learning is coupled with the EKF for the fine-tuning of the filter parameters with the objective of improving the estimation accuracy. The process noise covariance matrix was chosen for tuning, as it is crucial for calculating the filtering gain for the information contained in the system model and for the measurement to be utilized reasonably. In the Q-learning approach, the state s is related to the current value of the process noise covariance matrix. For example, it can be assigned as 0 for the nominal noise covariance matrix. The action a is related to the magnitude so as to enlarge or reduce the current process noise covariance matrix. It can be assigned as different numbers, such as 0, 1, 2 …, for different degrees of tuning. The reward R(s,a) is constructed based on the innovation of the EKF, which is a well-known indication of the estimation accuracy of the filter. Generally, a smaller innovation indicates a superior filtering performance, and vice versa.

For the system model shown in (23) and (38), the QLEKF algorithm is formulated in Algorithm 1. For simplicity, the subscript i of the weight vector $\Theta_{i}$, representing the number of iterations, is omitted here.



**
*Algorithm*
**
**
*1*
**
*: Q-learning extended Kalman filter*
1: ${\hat{x}}_{P,0}^{B}={\hat{x}}_{P,0}^{E}={\hat{x}}_{P,0}$, $P_{P,0}^{B}=P_{P,0}^{E}=P_{P,0}$                                                                    ⊳ Initialization2: k ← 03: Θ←0
4: **for** each period, **do**
5:     
**for** all a∈A, **do**
6:         
$C^{B}\leftarrow 0$, $C^{E}\leftarrow 0$

7:         
**for** t=1,2,⋯,T, **do**

8:             
k←k+1

9:             
$\left[{\hat{x}}_{P,k}^{B},P_{P,k}^{B},{\tilde{y}}_{P,k}^{B}\right]\leftarrow \mathrm{EKF}\left({\hat{x}}_{P,k-1}^{B},P_{P,k-1}^{B},y_{P,k},Q_{P,k},R_{P,k}\right)$                  ⊳ Benchmark filter
10:           
$\left[{\hat{x}}_{P,k}^{E},P_{P,k}^{E},{\tilde{y}}_{P,k}^{E}\right]\leftarrow \mathrm{EKF}\left({\hat{x}}_{P,k-1}^{E},P_{P,k-1}^{E},y_{P,k},{\hat{Q}}_{P,k}^{\left(s,a\right)},R_{P,k}\right)$                ⊳ Exploring filter
11:           
$\left[{\hat{x}}_{P,k},P_{P,k},{\tilde{y}}_{P,k}\right]\leftarrow \mathrm{EKF}\left({\hat{x}}_{P,k-1},P_{P,k-1},y_{P,k},{\hat{Q}}_{P,k}^{\left(s,a_{\max}\right)},R_{P,k}\right)$            ⊳ Main filter
12:             
$C^{B}\leftarrow C^{B}+\frac{1}{T-1}\left[{\left({\tilde{y}}_{k}^{B}\right)}^{T}R_{k}^{-1}{\tilde{y}}_{k}^{B}-C^{B}\right]$

13:             
$C^{E}\leftarrow C^{E}+\frac{1}{T-1}\left[{\left({\tilde{y}}_{k}^{E}\right)}^{T}R_{k}^{-1}{\tilde{y}}_{k}^{E}-C^{E}\right]$

14:           
**end for**
15:           
$R\left(s,a\right)\leftarrow C^{B}-C^{E}$                                                                            ⊳ Calculation of Reward 
16:           
$\Theta \leftarrow \Theta +\alpha \left[R\left(s,a\right)+\gamma \underset{a^{\prime }}{\max}\left(\Phi^{T}\left(s’,a’\right)\Theta \right)-\Phi^{T}\left(s,a\right)\Theta \right]\Phi \left(s,a\right)$     ⊳ Update of weight
17:           
${\hat{x}}_{P,k}^{E}\leftarrow {\hat{x}}_{P,k}^{B}$, $P_{P,k}^{E}\leftarrow P_{P,k}^{B}$                                                                   ⊳ Resetting of exploring filter
18:       
**end for**
19:       
$a_{\max}\leftarrow \arg\;\underset{a^{\prime }}{\max}\left(\Phi^{T}\left(s,a\right)\Theta \right)$                                                             ⊳ Selection of the best action20: **end for**21: **return** $\left\{{\hat{x}}_{P,k}\right\}$ and $\left\{P_{P,k}\right\}$                                                                                ⊳ Result of state estimation


In Algorithm 1, A is the pre-determined action space to be explored, T is a positive number representing the time interval for the calculation of the immediate reward R(s,a), ${\hat{x}}_{P,k}$ and $P_{P,k}$ are the state estimate and its corresponding error covariance matrix of the main filter, and ${\tilde{y}}_{P,k}$ is the innovation. Note that, in regard to the main filter, the process noise covariance matrix ${\hat{Q}}_{P,k}^{\left(s,a_{\max}\right)}$ tuned through Q-learning is used instead of $Q_{P,k}$ in the system model. 

It can be seen from the algorithm that there are another two parallel filters in addition to the main filter, where the benchmark filter and the exploring filter are distinguished by the superscripts B and S. The difference between the benchmark filter and the exploring filter is that the process noise covariance matrix of the benchmark filter $Q_{P,k}$ is obtained from the prior knowledge of the system, while that of the exploring filter ${\hat{Q}}_{P,k}^{\left(s,a\right)}$ is altered by enlarging or reducing $Q_{P,k}$.

The statistical values of the innovations obtained from the benchmark filter and the exploring filter are compared in order to construct the immediate reward R(s,a), which indicates whether the action a is preferable for improving the filtering performance. Note that the estimation of the exploring filter is reset after each update of the weight vector Θ to avoid the impacts of historical data on the evaluation of the action. An appropriate action $a_{\max}$ for tuning the process noise covariance matrix of the main filter is determined from the approximated Q function $\Phi^{T}\left(s,a\right)\Theta$. The output of the QLEKF is the estimation result of the main filter.

For clarity, the subroutine of the EKF is shown in Algorithm 2.
***Algorithm 2****: Extended Kalman filter*1: **function**
$\mathrm{EKF}\left({\hat{x}}_{P,k-1},P_{P,k-1},y_{P,k},Q_{P,k},R_{P,k}\right)$
$2\text{: }{\hat{x}}_{P,k|k-1}\leftarrow f\left({\hat{x}}_{P,k-1}\right)$                                                               ⊳ Prediction$3\text{: }P_{P,k|k-1}\leftarrow F_{P,k}P_{P,k-1}F_{P,k}^{T}+Q_{P,k}$$4\text{: }K_{P,k}\leftarrow P_{P,k|k-1}H_{P,k}^{T}{\left(H_{P,k}P_{P,k|k-1}H_{P,k}^{T}+R_{P,k}\right)}^{-1}$$5\text{: }{\tilde{y}}_{P,k}\leftarrow y_{P,k}-h\left({\hat{x}}_{P,k|k-1}\right)$$6\text{: }{\hat{x}}_{P,k}\leftarrow {\hat{x}}_{P,k|k-1}+K_{P,k}{\tilde{y}}_{P,k}$                                                     ⊳ Update$7\text{: }P_{P,k}\leftarrow \left(I-K_{P,k}H_{P,k}\right)P_{P,k|k-1}{\left(I-K_{P,k}H_{P,k}\right)}^{T}+K_{P,k}R_{P,k}K_{P,k}^{T}$8:      **return**
${\hat{x}}_{P,k}$$,\;P_{P,k}$$,\;{\tilde{y}}_{P,k}$
9: **end function**

In Algorithm 2, $K_{P,k}$ denotes the Kalman gain. It is easy to see from the algorithm that the EKF is designed based on the system model shown in Section 2.3. The innovation ${\tilde{y}}_{P,k}$, as the output of Algorithm 2, is not used in the next iteration, but it is used to construct the immediate reward R(s,a) in Algorithm 1.

## 5. Simulations

### 5.1. Simulation Conditions

Simulations were performed to evaluate the performance of the space-target-based INS/Vision integrated navigation. An ideal trajectory was produced for the unmanned aircraft with an altitude of about 30 km and velocity of about 680 m/s, including the true position, velocity and attitude. The noises of the IMU and the star camera were added to the ideal values to obtain the simulated measurement data. For the data generation of the gyroscope, the angle random walk, the rate random walk and the bias were taken into consideration. The LEO satellites (low Earth orbit) with a semi-major axis of 6938 km and inclination of 55° were seen as the observed space targets. The construction of the satellite constellation is shown in Figure 7.

The parameters for the integrated navigation simulation are listed in Table 1.

It should be pointed out that the estimation performance and computational effort of the algorithm depend on the QLEKF parameters. In particular, it can be seen from the simulation results that the QLEKF parameters shown in Table 1 yield an improved performance, and the convenience of calculation is retained. Further works are planned in which we will perform a rigorous comparison analysis for the estimation performance using different QLEKF parameters. The absolute position, velocity and attitude of the aircraft are estimated with the method shown in Section 2 and Section 4 in cases where the positions of the space targets are known. The comparison between the state estimation and the ideal trajectory yields the navigation results.

### 5.2. CRLB Based on the Visibility Analysis

The visibility of the space targets for the star camera on the aircraft is analyzed using the principles shown in Section 3. The curves of the apparent magnitude of some space targets over time are shown in Figure 8. It can be seen from the figure that, as the specified sensitivity of the star camera in Algorithm 1 is 8 Mv, some space targets are bright enough to be observed in certain areas of the orbit.

The curves of the Sun-Earth-target angles of some space targets over time are shown in Figure 9. The angle bound $90{}^{\circ}+\beta^{\left(i\right)}$, representing Earth’s shadow, is plotted as the shadow in the figure. From this result, we can see that the space targets are not in Earth’s shadow for most of the time.

The curves of the Earth-aircraft-target angles, Sun-aircraft-target angles of some space targets over time and the corresponding angle bounds are shown in Figure 10 and Figure 11, respectively. The figures illustrate that, sometimes, the space targets are not occulted by the Earth or Sun. 

Based on the previous analysis, the potential performance of the presented navigation method was evaluated through the CRLB, which was derived from the system model. The CRLB provides a theoretical bound on the achievable accuracy of the navigation system. Thus, it is useful to study the feasibility of a novel navigation scheme. The calculation procedure of the CRLB can be found in the literature [36,37].

Figure 12 and Figure 13 describe the theoretical bounds of the three-axis position estimation errors of the integrated navigation system with different noise levels of the accelerometer and the star camera in cases where only the visible space targets are observed. It indicates that a high navigation performance is achievable using the state-of-the-art sensors by applying the space-target-based INS/Vision integrated navigation method.

### 5.3. Simulation Results of the Navigation Filter

The navigation results of the space-target-based INS/Vision integration analysis obtained from the EKF are shown in Figure 14, Figure 15 and Figure 16, including the curves of the position, velocity and attitude errors, where the solid lines represent the estimation error and the dashed lines represent the 3σ bound, calculated from the error covariance matrix of the EKF.

As expected, the position, velocity and attitude can all be estimated correctly by the measurement of the LOS vectors of the space targets and starlight vectors. The mean estimation errors of the position, velocity and attitude of the space-target-based INS/Vision integration were 12.96 m, 0.32 m/s and 0.94″, respectively. This shows that the presented method has the potential to realize the absolute navigation of unmanned aircrafts with a high accuracy. For the implementation of the navigation method, the position error of the space target should be sufficiently less than the accuracy specification of the unmanned aircraft. In the simulation, only the measurements of the visible space targets are used to update the state vector. In the case where there are no visible space targets, the state vector is predicted by the IMU. The effect of the visibility is not distinct, as there are multiple space targets, as shown in Figure 7.

Next, the ability of the QLEKF to improve the navigation accuracy for aircraft navigation through space-target-based INS/Vision integration was assessed. Monte Carlo simulations were conducted to evaluate the performance of the QLEKF in comparison with the traditional EKF and adaptive extended Kalman filter (AEKF) [38]. The QLEKF aims to fine-tune the noise covariance matrix obtained from the prior knowledge, while the AEKF aims to estimate the noise covariance matrix together with the state vector. Figure 17 and Figure 18 show the norms of the root mean squared errors (RMSE) of the position and attitude estimates obtained from the EKF, the AEKF and the QLEKF. 

According to the simulation results, the EKF provides similar results to the AEKF, while the QLEKF outperforms both the EKF and the AEKF with higher precision on average. These results demonstrate that the Q-learning approach is valuable for fine-tuning the critical parameters of the navigation filter.

## 6. Conclusions

This paper presents a novel navigation method that can be used to estimate the position, along with velocity and attitude, of unmanned aircrafts, using the star camera to observe the LOS vectors of the space targets with known positions. This method is especially suitable for aircrafts with long-term and high-accuracy autonomous navigation requirement. Based on the mathematical model and the visibility analysis, the CRLB of the integrated navigation system was calculated in order to validate the effectiveness of the presented method. In addition, the QLEKF was designed to enhance the navigation performance. The simulation results demonstrate the high performance of the space-target-based INS/Vision integrated navigation method. Further research will focus on the improvement of the navigation scheme in order to address the unfavorable effects of the space target position error and optical measurement error caused by thermal environment on the performance of the system.

## Figures and Tables

**Figure 1 sensors-22-06992-f001:**
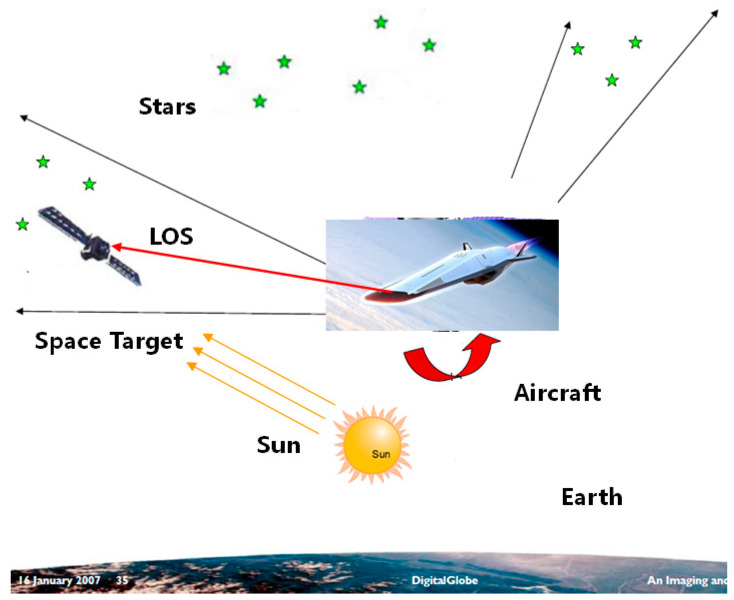
The measurement of the space target LOS vector.

**Figure 2 sensors-22-06992-f002:**
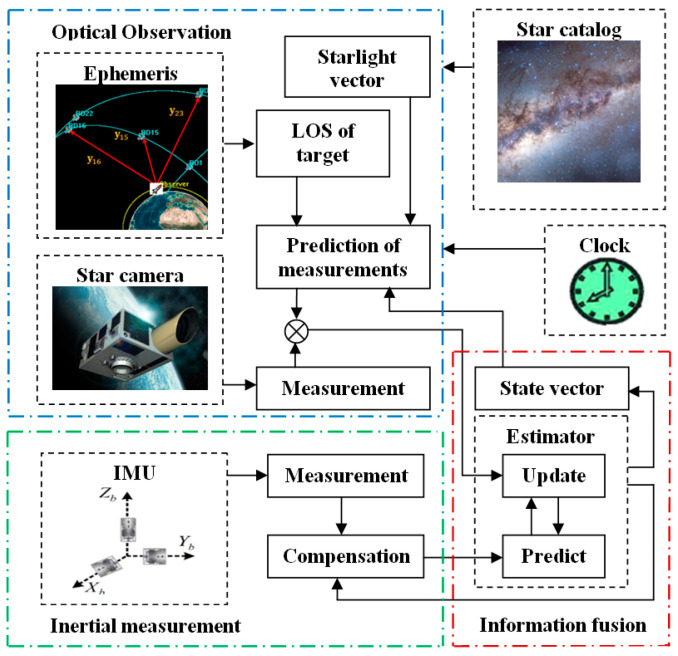
Space-target-based INS/Vision integrated navigation method.

**Figure 3 sensors-22-06992-f003:**
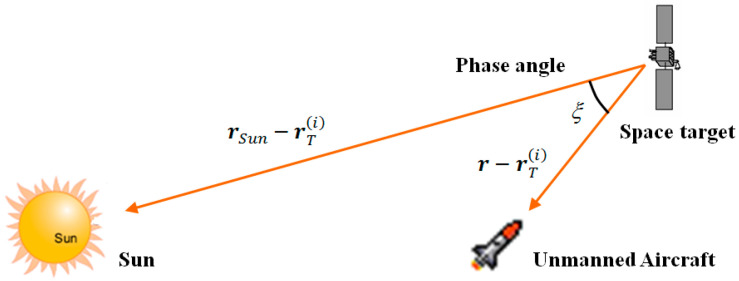
Relative positions of the Sun, space target and unmanned aircraft.

**Figure 4 sensors-22-06992-f004:**
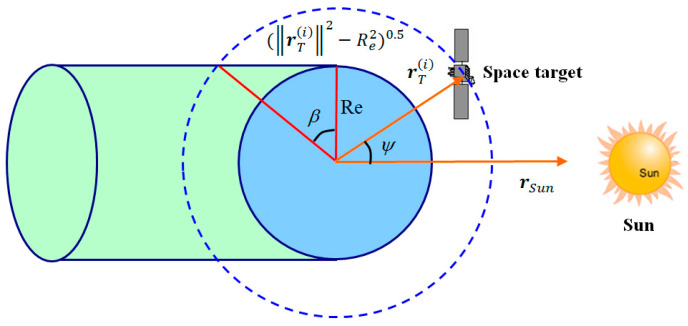
Earth’s shadow in the orbit of space target.

**Figure 5 sensors-22-06992-f005:**
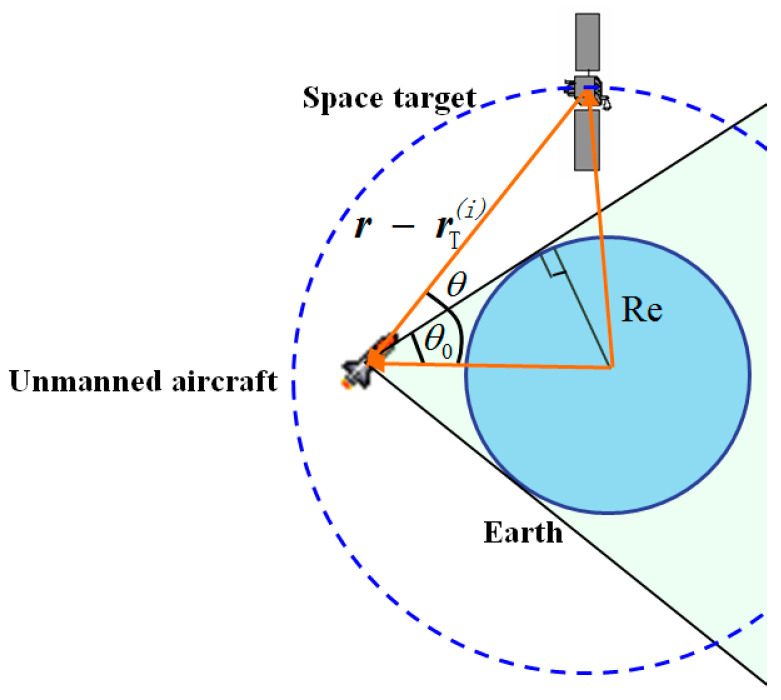
Relative positions of the Earth, aircraft and target.

**Figure 6 sensors-22-06992-f006:**
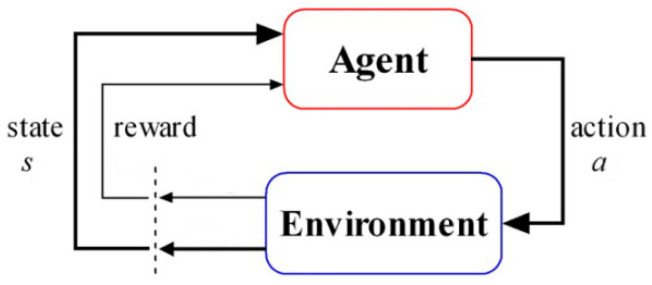
Main components of the iterative learning procedure.

**Figure 7 sensors-22-06992-f007:**
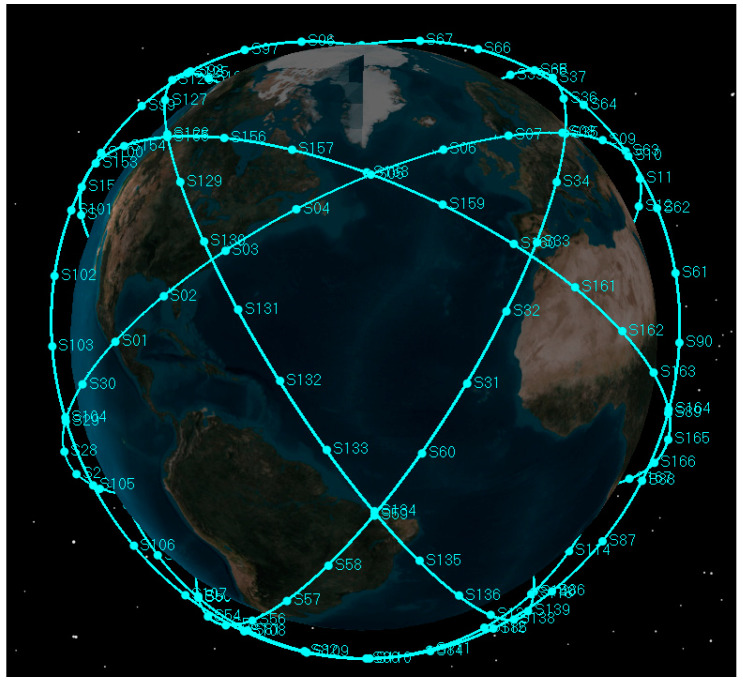
Satellite constellation in the simulation scenario.

**Figure 8 sensors-22-06992-f008:**
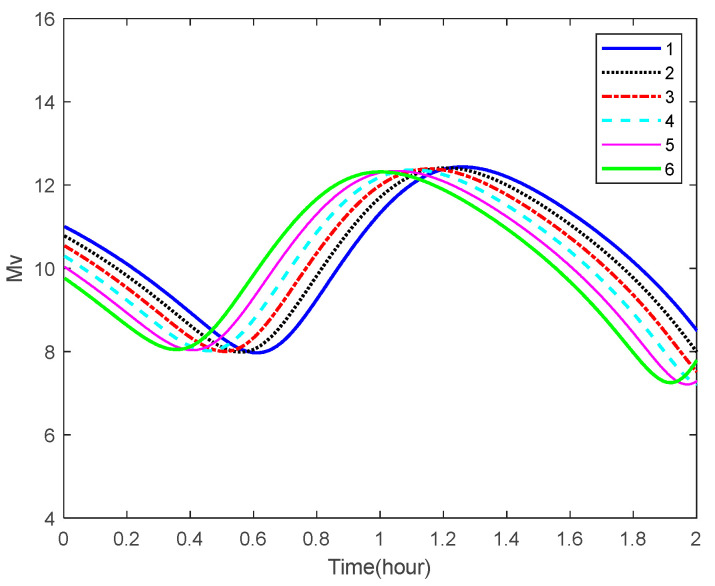
Apparent magnitudes of the space targets.

**Figure 9 sensors-22-06992-f009:**
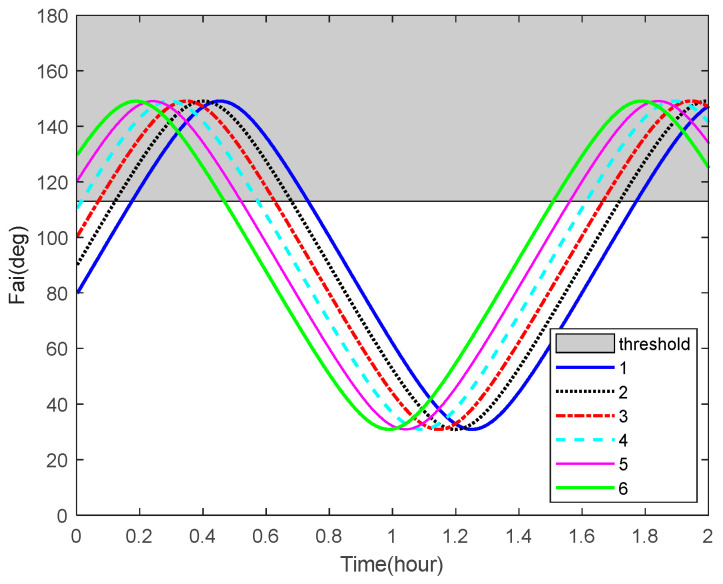
Sun–Earth–target angles and the corresponding bounds.

**Figure 10 sensors-22-06992-f010:**
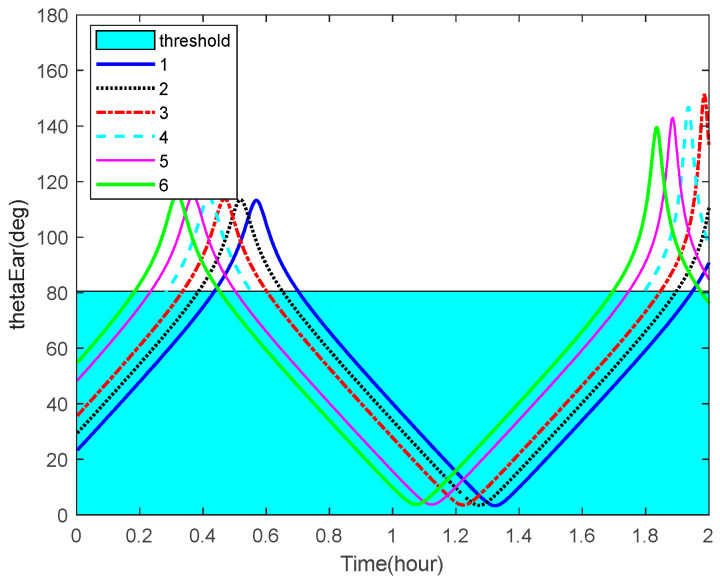
Earth-aircraft-target angles and the corresponding bounds.

**Figure 11 sensors-22-06992-f011:**
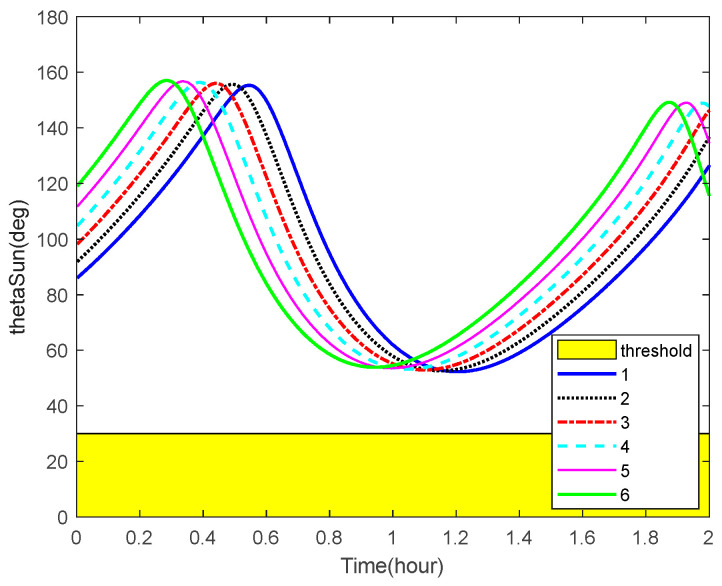
Sun-aircraft-target angles and the corresponding bounds.

**Figure 12 sensors-22-06992-f012:**
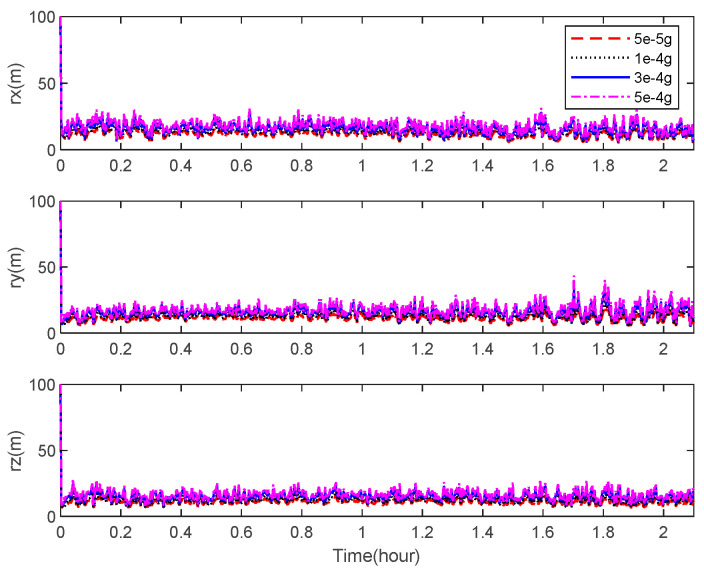
CRLB for the position estimation with different accelerometer noise levels.

**Figure 13 sensors-22-06992-f013:**
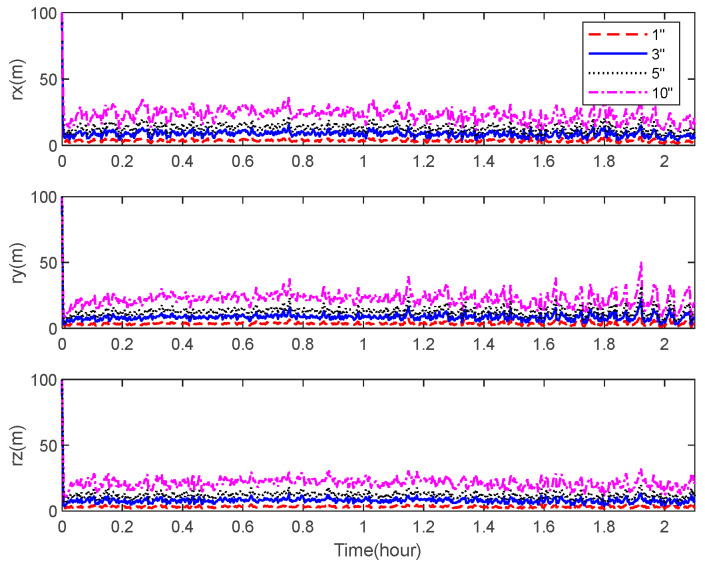
CRLB for the position estimation with different star camera noise levels.

**Figure 14 sensors-22-06992-f014:**
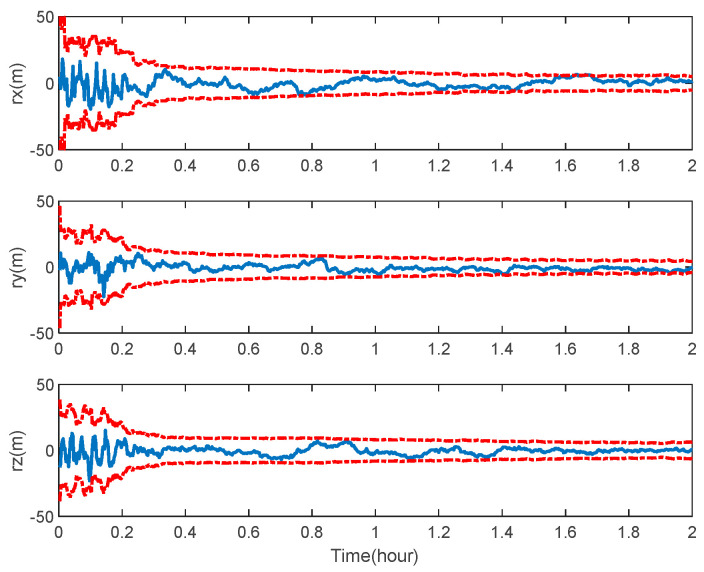
Position error of space-target-based INS/Vision integrated navigation.

**Figure 15 sensors-22-06992-f015:**
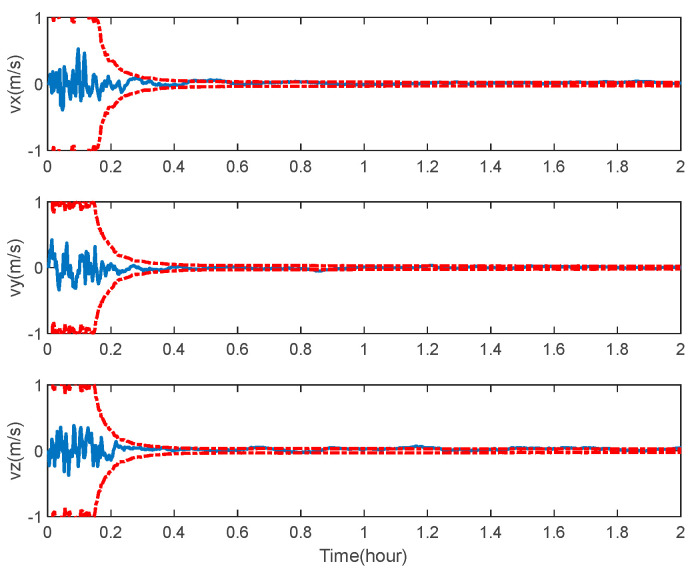
Velocity error of space-target-based INS/Vision integrated navigation.

**Figure 16 sensors-22-06992-f016:**
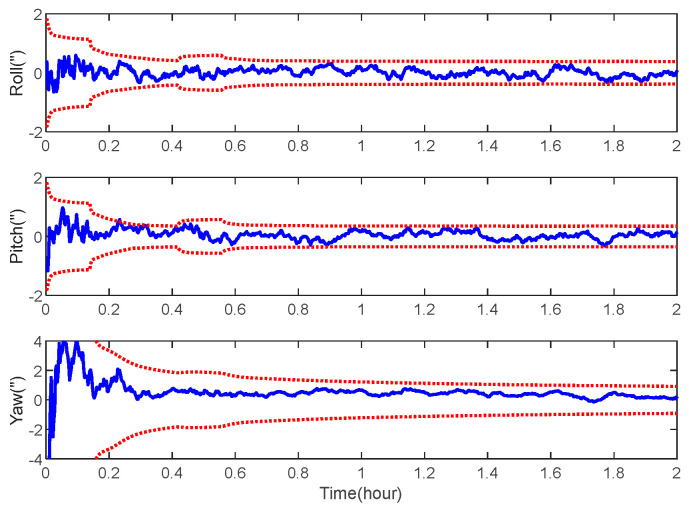
Attitude error of space-target-based INS/Vision integrated navigation.

**Figure 17 sensors-22-06992-f017:**
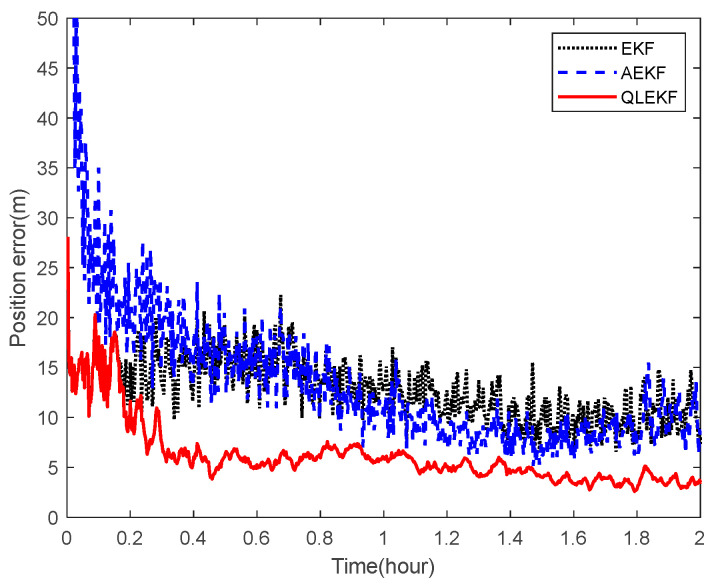
RMSE of the position estimation for EKF, AEKF and QLEKF.

**Figure 18 sensors-22-06992-f018:**
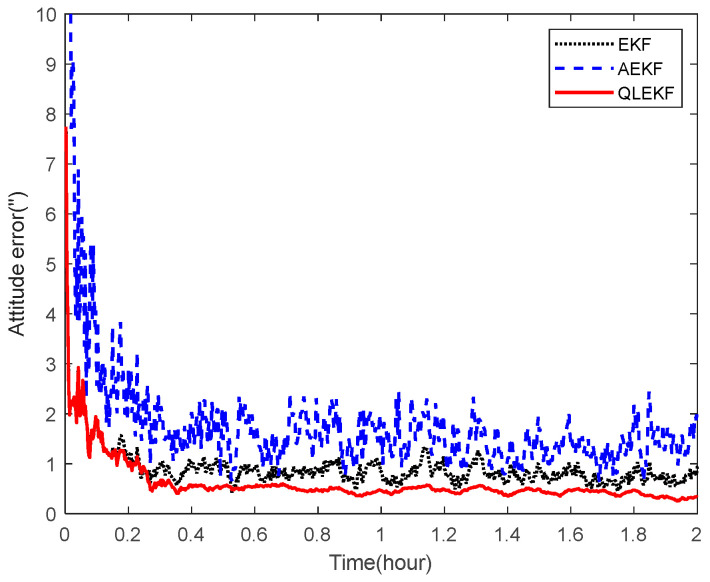
RMSE of the attitude estimation for EKF, AEKF and QLEKF.

**Table 1 sensors-22-06992-t001:** Basic simulation configuration.

Initial condition	Initial position error	(200 m, 200 m, 200 m)
	Initial velocity error	(0.02 m/s, 0.02 m/s, 0.02 m/s)
	Initial attitude error	(0.01°, 0.01°, 0.01°)
	Duration of navigation process	2 h
Measurement performance	Accuracy of gyroscope	0.02°/hour
	Accuracy of accelerometer	100 μg
	Accuracy of star camera	5″
	Sensitivity of star camera	8 Mv
	Measurement update frequency	1 Hz
	Observed target in each period	1
	Total number of space target	180
QLEKF parameter	Learning rate	0.1
	Discounter factor	0.9
	Parameter of basis function $\mu_{ad}$	(−2, −1, 0, 1, 2)
	Parameter of basis function $\sigma_{ad}$	0.3
	Tuning of filtering parameter	${\hat{Q}}_{P,k}^{\left(s,a\right)}=10^{2a}Q_{P,k}$

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
