# Peer review of "Autonomous Navigation of Unmanned Aircraft Using Space Target LOS Measurements and QLEKF"

_sensors, 2022, doi:10.3390/s22186992_

Round 1

Reviewer 1 Report

This paper presents a novel navigation method to estimate position along with locity and attitude for unmanned aircrafts using the star camera, a QLEKF is designed for the performance enhancement of the integrated navigation system. The structure of the article is reasonable, the expression is clear, and the content of the article can generally support the author's innovation.

Author Response

Dear Editor:

The paper entitled: “Autonomous Navigation for Unmanned Aircraft Using Space Target LOS Measurement and QLEKF” with Manuscript ID sensors-1875487 has been revised according to the reviews’ comments. A summary of the changes that we have made is as follows:

Reply to comments of reviewer #1:

  1. Comment:

This paper presents a novel navigation method to estimate position along with locity and attitude for unmanned aircrafts using the star camera, a QLEKF is designed for the performance enhancement of the integrated navigation system. The structure of the article is reasonable, the expression is clear, and the content of the article can generally support the author's innovation.

Reply: Thanks.

Sincerely,

XIONG Kai

Science and Technology on Space Intelligent Control Laboratory,

Beijing Institute of Control Engineering, Beijing 100094, China

E-mail: tobelove2001@vip.tom.com

Reviewer 2 Report

Dear Authors,

your manuscript deals with the interesting approach to fuse celestial Navigation with INS. In detail you integrate star sensor measurements to LEOs with INS measurements. Although I am not totally convinced about the practical implementation of your approach I really appreciated your noted mathematical model (tuning the system noise via QLEKF).

please find below a number of technical hints/questions followed by a

short section on grammar hints. My comments refer to the line numbering scheme of the manuscript

A) technical comments

1) l43; make clear in the introduction if your star sensor is just used to track LEOs or if your approach is also suited to monitor star positions (at last during night time); in figure 1 both stars and satellites are intended targets-

but how to deal with stars during day time? In L155 you definitely refer to stars as targets

2) as in your example /simulations in chapter 5 just satellites are the targets

I recommend to add at least in chapter 2 a comment which brightness and number of stars shall be implemented (number of stars, quality of position, magnitude)

3) l 248 -252

I wonder slightly about the equations (42) and (43)

In (43) you calculate the absolute magnitude of the space target;

according to the text 'rad' denotes the radius ( do you mean here the semi-major axis of the satellite ??) and according to the text 'a' stands for reflectivity. Do you mean here 'albedo' or anything else? specify the

range this quantity a may have?

Moreover its quite dubious as in most literature a=semi -major axis or radius of the satellite orbit; think about some re-naming of the quantities

In formula (42) I wonder about the outcome of the log term; In my calculations the quantity in brackets is always small than one and therefore the log negative-this leads to a brighter apparent magnitude of the satellite than its absolute magnitude (but should be vice versa). Please check once more formula (42)

4) I wonder about the text in line 274:

' it should not be occulted by celestial bodies, such as Earth, Sun and Moon.'

Its clear that the target satellite or star shall not be occulted by the Earth. But at least target satellites cannot be occulted by Moon and Sun ??? or do I misinterpret here the text? make clear to the readers

5) the figure 5 should be improved; although its just a sketch but the sketch gives the impression that the unmanned aircraft flies in a higher altitude as the satellite

6) l320 , formula (54)

  please give an explanation or just a reference why you use especially this function as the basis function

7) l397

here you note 'The noises of the IMU and the star camera are added'

specify in more detail; do you just enter white noise for both sensors or at

least coloured noise and/or biased for the IMU?

8) Table 3 , QLEKF

explain to the readers why you have chosen especially these QLEKF parameters; from test calculations? from literature? just by chance ?

9) coordinates of space targets

which restrictions do you have for the quality of the space target position?

do you somehow simulate also epochs where you star sensor has to switch from target1 satellite to another target satellite? are there epochs (how many) without star sensor observations? how does this impact the state vector?

10) Appendix

Is the proof of the theorem really necessary in this full length?

B) grammar hints

1) l49

 '.. also suffers from an increasing error...'

2) l57

'.. can be extracted directly using...'

3) l66

'.. to achieve a higher accuracy than...'

4) l111

'.. to generate the attitude solution...'

5) l120

'.. represents the Euclidean norm.'

6) l126

'.. represents the quaternion product...'

7) l153

'.. the following function of the quaternion is...'

8) l170

'From (16) and (17) we obtain'

9) l181

'.. the state vector x(A,k) by means of  the ...'

10) l191

'.. established to derive the dynamics...'

11) l324

'.. a<1 is the learning rate ... is the discount factor...'

12) l345

'.. of the theorem is provided in Appendix A. It ...'

13) l388

'.. In Algorithm 2, K(P,k) denotes the Kalman gain.'

14) l403 , just above table 3

'.. are listed in Table ?' I assume this should read '3' and not '1' as in the text

15) l420

'.. in Earth's shadow most of the time.'

best regards

Author Response

Dear Editor:

The paper entitled: “Autonomous Navigation for Unmanned Aircraft Using Space Target LOS Measurement and QLEKF” with Manuscript ID sensors-1875487 has been revised according to the reviews’ comments. A summary of the changes that we have made is as follows:

Reply to comments of reviewer #2:

  1. Comment:

l43; make clear in the introduction if your star sensor is just used to track LEOs or if your approach is also suited to monitor star positions (at last during night time); in figure 1 both stars and satellites are intended targets-but how to deal with stars during day time?

Reply: Some explanations about the daytime star camera are added in the revised paper (line.10, page.2): “Different from the traditional star sensor, the star camera images both the low-orbit satellites and the background stars in the field-of-view (FOV), and the LOS vectors of the satellites in inertial frame can be extract directly using the star catalogue and the least square (LS) algorithm. The current star cameras have the ability to observe the low-orbit satellites and the stars simultaneously during daytime for unmanned aircrafts in near space”.

  1. Comment:

In L155 you definitely refer to stars as targets as in your example /simulations in chapter 5 just satellites are the targets. I recommend to add at least in chapter 2 a comment which brightness and number of stars shall be implemented (number of stars, quality of position, magnitude)

Reply: Some remarks about the observed stars are added in the revised paper (line.1, page.3): “It should be mentioned that, the starlight vectors of the stars in the inertial frame can be calculated with the accuracy of less than 1 mas. The number and magnitude of the observed stars depends on the FOV and the sensitivity of the star camera. Generally, a star camera with the FOV on the order of  and the sensitivity of about 8 Mv is feasible for navigation”.

  1. Comment:

l 248 -252

I wonder slightly about the equations (42) and (43). In (43) you calculate the absolute magnitude of the space target according to the text 'rad' denotes the radius ( do you mean here the semi-major axis of the satellite ??) and according to the text 'a' stands for reflectivity. Do you mean here 'albedo' or anything else? specify the range this quantity a may have? Moreover its quite dubious as in most literature a=semi -major axis or radius of the satellite orbit; think about some re-naming of the quantities. In formula (42) I wonder about the outcome of the log term; In my calculations the quantity in brackets is always small than one and therefore the log negative-this leads to a brighter apparent magnitude of the satellite than its absolute magnitude (but should be vice versa). Please check once more formula (42).

Reply: In the revised paper, some quantities are renamed and equation (43) is rewritten as (line.19, page.7)

where  is the apparent magnitude of Sun,  is the observed body radius, and  is the body albedo of the space target,  is an astronomical unit. In addition, Equation (42) is checked considering that the definition of the absolute magnitude for space targets is different from that for stars.

  1. Comment:

I wonder about the text in line 274: ' it should not be occulted by celestial bodies, such as Earth, Sun and Moon.' Its clear that the target satellite or star shall not be occulted by the Earth. But at least target satellites cannot be occulted by Moon and Sun ??? or do I misinterpret here the text? make clear to the readers.

Reply: Some explanations about the occultation of the space target are added in the revised paper (line.8, page.8): “A space target is believed to be occulted when its LOS vector points to a celestial body. Generally, the LOS vector of the space target cannot be measured effectively by the star camera in this case as the images of the space target and the celestial body are overlapped”.

  1. Comment:

the figure 5 should be improved; although its just a sketch but the sketch gives the impression that the unmanned aircraft flies in a higher altitude as the satellite

Reply: In the revised paper (line.13, page.8), figure 5 is modified to avoid confusion.

  1. Comment:

l320 , formula (54)

  please give an explanation or just a reference why you use especially this function as the basis function

Reply: Some explanations about the basis function is added in the revised paper (line.35, page.9): “It should be mentioned that the choice of the basis function is up to the user and a bad choice may lead to poor filtering performance. In this study, the basis function is designed with the structure similar to [38], such that the appropriate action can be activated”.

  1. Comment:

l397

here you note 'The noises of the IMU and the star camera are added' specify in more detail; do you just enter white noise for both sensors or at least coloured noise and/or biased for the IMU?

Reply: Some explanations about the measurement data generation are added in the revised paper (line., page.): “For data generation of the gyroscope, the angle random walk, the rate random walk and the bias are taken into consideration”.

  1. Comment:

Table 3 , QLEKF

explain to the readers why you have chosen especially these QLEKF parameters; from test calculations? from literature? just by chance ?

Reply: Some remarks about the QLEKF parameters are added in the revised paper (line.11, page.12): “It should be pointed out that the estimation performance and computational effort of the algorithm depend on the QLEKF parameters. Particularly, it can be seen from the simulation results that the QLEKF parameters shown in Table 3 yield an improved performance, and the convenience of calculation is retained. Further works are planned to make a rigorous comparison analysis for the estimation performance with different QLEKF parameters”.

  1. Comment:

coordinates of space targets

which restrictions do you have for the quality of the space target position? do you somehow simulate also epochs where you star sensor has to switch from target1 satellite to another target satellite? are there epochs (how many) without star sensor observations? how does this impact the state vector?

Reply: Some remarks about the implementation of the navigation method is added in the revised paper (the last line in page.16): “For the implementation of the navigation method, the position error of the space target should be sufficiently less than the accuracy specification of the unmanned aircraft. In the simulation, only the measurements of the visible space targets are used to update the state vector. In the case that there are no visible space targets, the state vector is predicted with the IMU. The effect of the visibility is not distinct as there are multiple space targets as shown in Figure 7”.

  1. Comment:

Is the proof of the theorem really necessary in this full length?

Reply: The proof of the theorem may be helpful for readers to comprehend the idea of the learning procedure.

  1. Comment:

grammar hints

        Reply: We have done our best to fix the typos and grammatical errors.

Sincerely,

XIONG Kai

Science and Technology on Space Intelligent Control Laboratory,

Beijing Institute of Control Engineering, Beijing 100094, China

E-mail: tobelove2001@vip.tom.com

Reviewer 3 Report

The manuscript: “Autonomous Navigation for Unmanned Aircraft Using Space Target LOS Measurement and QLEKF”

I think the paper is of high quality. I recommend publishing with Sensors.

I wonder what the differences are between the Q-learning and adaptive filters, and how the Q-learning methods are applied in the other fields.

Author Response

Dear Editor:

The paper entitled: “Autonomous Navigation for Unmanned Aircraft Using Space Target LOS Measurement and QLEKF” with Manuscript ID sensors-1875487 has been revised according to the reviews’ comments. A summary of the changes that we have made is as follows:

Reply to comments of reviewer #3:

  1. Comment:

I think the paper is of high quality. I recommend publishing with Sensors. I wonder what the differences are between the Q-learning and adaptive filters, and how the Q-learning methods are applied in the other fields.

Reply: Thanks. Some explanations about the difference between the QLEKF and the AEKF are added in the revised paper (line.9, page.17): “The QLEKF aims to fine tune the noise covariance matrix obtained from the prior knowledge, while the AEKF is designed to estimate the noise covariance matrix together with the state vector”. The application of the Q-learning methods is briefly described in Section 4 (line.15, page.9): “Q-learning has received considerable attention, with many successful applications in various fields such as game, robots, industrial process control and network management”.

Sincerely,

XIONG Kai

Science and Technology on Space Intelligent Control Laboratory,

Beijing Institute of Control Engineering, Beijing 100094, China

E-mail: tobelove2001@vip.tom.com

Author Response

Dear Editor:

The paper entitled: “Autonomous Navigation for Unmanned Aircraft Using Space Target LOS Measurement and QLEKF” with Manuscript ID sensors-1875487 has been revised according to the reviews’ comments. A summary of the changes that we have made is as follows:

Reply to comments of reviewer #4:

  1. Comment:

This paper presents a navigation scheme by introducing the observation of a space target, i.e. Line-of-sight vector to integrate with IMU in the GNSS-denied environment. It is said that the ephemeris of the space targets is known. There is a problem with this assumption. How to update the ephemeris? It is not similar to the scheme of GNSS, of which the ephemeris is updated by the ground station and can be broadcasted to the receivers. Then, another question is how to update it in the denied environment.

Reply: Some explanations about the implementation of the navigation method in the denied environment is added in the revised paper (the first line in page.4): “For the implementation of the navigation method, multiple space targets should be tracked by the ground stations, and the ephemeris of the space targets is updated before the unmanned aircraft takes off. Then the positions of the space targets can be obtained through the orbital propagation during the flight of the aircraft”.

  1. Comment:

Since the image process cost time and the velocity of the aircraft is quite fast, the navigation solution is delayed. So how guarantee the real-time accuracy?

Reply: Some explanations about the data processing of the navigation method is added in the revised paper (line.16, page.3): “The navigation filter, which is designed based on the state model and the observation model of the navigation system, is implemented to update the prediction of the position, velocity and attitude based on the measurements from the star camera. As the position, velocity and attitude of the unmanned aircraft are predicted with the IMU with high data update rate, the real-time accuracy can be guaranteed in the case that the exposure time of the star camera is recorded accurately”.

  1. Comment:

Line 356-358 in page 10, suggest the authors detail the relationship between s and process noise covariance matrix, and give a specific definition of a.

Reply: Some explanations about the state and the action are added in the revised paper (line.33, page.10): “In the Q-learning approach, the state s is related to the current value of the process noise covariance matrix. For example, it can be assigned as 0 for the nominal noise covariance matrix. The action a is related to the magnitude to enlarge or reduce the current process noise covariance matrix. It can be assigned as different numbers, such as 0, 1, 2 …, for different degree of the tuning”.

  1. Comment:

Figure 9 to Figure 11 show that the visibility is not satisfied for some time, suggest the authors carry out the corresponding experiments in this case.

Reply: Some explanations about the effect of the visibility are added in the revised paper (line.2, page.17): “In the simulation, only the measurements of the visible space targets are used to update the state vector. In the case that there are no visible space targets, the state vector is predicted with the IMU. The effect of the visibility is not distinct as there are multiple space targets as shown in Figure 7”.

  1. Comment:

From page 11 to page 11, consider replacing the superscript of angle, “o” with “o”.

Reply: In the revised paper (line.14, page.12), the superscripts of angles are modified.

  1. Comment:

Line 520 in page 18, consider replacing “i=0” with “i=1”.

Reply: In the revised paper (the last line in page.18), this error is fixed.

  1. Comment:

In equation (25), consider replacing the subscript of Q, “∞” with “i”.

        Reply: The equation (A25) has been checked.

Finally, we would thank the reviewers and the co-editor for their valuable comments which help us to improve the quality of our paper. We appreciate your consideration for publication in Sensors.

Look forward to receiving a favorable reply from you soon.

Sincerely,

XIONG Kai

Science and Technology on Space Intelligent Control Laboratory,

Beijing Institute of Control Engineering, Beijing 100094, China

E-mail: tobelove2001@vip.tom.com

Round 2

Reviewer 4 Report

The authors have solved all of my comments. I recommend it accepted.